

# Identification of Snowfall Riming and Aggregation Processes Using Ground-Based Triple-Frequency Radar

Wang Danyang [1,2], He Wenying [1,2,*], Bi Yongheng [1,2], Xia Xiangao [1,2], Chen Hongbin [1,2]

[1] Laboratory of Middle Atmosphere and Global Environment Observation,The Institute of Atmospheric Physics,Chinese Academy of Sciences, Beijing, China
[2] University of Chinese Academy of Sciences, Beijing, China

*Correspondence to*: He Wenying (hwy@mail.iap.ac.cn)





**Abstract.** Riming and aggregation are critical ice-phase microphysical processes in winter clouds, but their overlapping signatures and dynamic transitions pose challenges for conventional single-frequency radar detection. We introduce a novel gradient-based identification method using ground-based triple-frequency dual-polarization radar observations. By analyzing vertical gradients of triple-frequency radar variables, rather than their absolute values, we discern these
microphysical processes through physically based thresholds that reflect particle growth regimes. This approach captures subtle spatiotemporal variations in riming and aggregation that conventional threshold methods would miss, particularly in resolving layered riming-aggregation transitions. The dynamic gradient-based method demonstrates the enhanced physical consistency and adaptability near process boundaries, which obviously improve the tracking of ice-particle evolution. These advances provide a pathway to refine microphysical parameterizations and enhance high-resolution snowfall forecasting.

**1 Introduction**

Ice-phase microphysical processes in winter snowfall clouds, particularly riming and aggregation as the critical ice particle growth mechanisms, significantly influence snowfall formation and prediction. However, substantial uncertainties remain in current understanding and quantitative characterization of these processes (Oue et al., 2021). Riming refers to the process where super-cooled cloud droplets rapidly freeze upon collision with ice crystals or snowflakes and adhere to their surfaces,
while aggregation describes the microphysical process where ice crystals collide and bond within clouds to form larger snowflakes or snow clusters. The two processes markedly alter the size, shape, and density of snow particles, thereby affecting snowfall intensity and even precipitation phase. In winter snowfall clouds, riming and aggregation often occur simultaneously alongside other processes such as sedimentation and fragmentation. These complex and variable microphysical processes are difficult to effectively distinguish using single-band radar observations alone. The
parameterization of ice-phase processes in numerical models remains inadequate, leading to significant uncertainties in snowfall forecasts. Therefore, utilizing advanced radar observations to identify the evolution of riming and aggregation processes in snowfall clouds is of great scientific significance for improving the understand of ice-phase microphysics and enhancing snowfall prediction accuracy.

Advances in radar technology have enabled new possibilities for investigating the microphysical mechanisms of snowfall
formation through dual- and triple-frequency radar observations. By systematically analyzing the scattering and absorption differences of hydrometeors at different frequencies and comprehensively utilizing radar observation parameters, including radar reflectivity (Ze), mean Doppler velocity (MDV), spectral width (SW), and linear depolarization ratio (LDR), multi-dimensional microphysical information such as particle size, density, phase, and shape can be effectively retrieved. Newly rimed particles are compact and nearly spherical (2-5 mm) with high density. In contrast, aggregated particles are formed by
the collision and combination g of multiple ice crystals, exhibiting characteristics such as larger sizes (up to ~20 mm), lower



density, loose structure, and irregular shape (Braham, 1990; Bringi et al., 2017; Teisseire et al., 2025; Von Terzi et al., 2022). These microphysical characteristics lead to pronounced differences in radar observations between riming- and aggregation-dominated regions. In riming-dominated areas with abundant liquid water, Ze can reach up to 25 dBZ due to the presence of graupels or dense snow particles (Bringi et al., 2017), whereas in aggregation-dominated regions, owing to ice's lower

dielectric constant and the porous nature of aggregates, Ze typically remains between 0 and 15 dBZ (Kneifel et al., 2015; Matrosov et al., 2019). MDV, representing the mean fall velocity, and SW, indicating the spread of particle velocities, are closely related to particle size, density, and shape, making them widely useful for particle identification (Grazioli et al., 2015a; Kneifel and Moisseev, 2020; Mason et al., 2018; Mosimann, 1995). Rimed particles, being denser and more spherical—and often occurring in dynamically active regions (e.g., convection or strong downdrafts)—exhibit higher MDV

(>1.5 m/s) and SW compared to aggregated particles (Grazioli et al., 2015b; Vogl et al., 2022). Aggregation processes primarily occur in relatively stable regions with more uniform particle types, thus displaying lower SW (<0.3 m/s) and MDV (0.5–1.2 m/s). LDR, which measures particle non-sphericity and orientation consistency, is approximately –15 dB for aggregated snowflakes and can increase to around –10 dB near melting (Teisseire et al., 2025; Vogl et al., 2022). In contrast, rimed particles, being nearly spherical with smooth ice coatings and high symmetry, yield virtually no cross-polarized return,

resulting in LDR values between –36 and –28 dB (Bringi et al., 2017; Tyynelä and Chandrasekar, 2014).

Beyond direct multi-frequency radar reflectivity measurements, dual-wavelength ratios (DWR) provide a new perspective for identifying riming and aggregation. When particle diameters are much smaller than radar wavelengths, all frequencies operate in the Rayleigh scattering regime, and DWR differences are minimal. For rimed particles (~2–5 mm), the W-band wavelength (~3 mm) is comparable to particle size, causing non-Rayleigh scattering (i.e., Ze is non-monotonic with particle

size); at Ka-band (~8.6 mm), scattering lies in the transition between Rayleigh and Mie regimes; while at X-band (3 cm), particles still satisfy the Rayleigh approximation (Ze $\propto$ D6). Consequently, riming leads to significant increases in $DWR_{Ka-W}$ but only modest increases in $DWR_{X-Ka}$ (Dias Neto et al., 2019; Kneifel et al., 2015; Mason et al., 2018; Tridon et al., 2022). For larger aggregated particles (~5–20 mm), both W- and Ka-band observations fall within the Mie regime—with Ze oscillating rather than increasing monotonically with size—causing $DWR_{Ka-W}$ to saturate at roughly 7–10 dB (Dias Neto et

al., 2019). However, at X-band, where scattering transitions from Rayleigh to Mie for these larger particles, Ze continues to grow significantly with aggregate size, driving continued increases in $DWR_{X-Ka}$ and producing the characteristic "triple-frequency DWR hook" pattern (Kneifel et al., 2015; Leinonen et al., 2012; Mason et al., 2019).

Given that the vertical gradients of radar observables contain rich information about precipitating particle evolution, Planat et al., (2021) innovatively proposed a process identification method based on vertical gradient signatures (PIVS) using

single-frequency radar Ze and its polarimetric profiles. Their results demonstrated that this method could effectively identify regions where riming and aggregation processes coexist. Although single-frequency radar PIVS has limitations in quantitatively distinguishing dominant processes, it provided an important foundation for developing more precise multi-parameter retrieval methods.





Precipitation particles in winter clouds exhibit pronounced spatiotemporal variability in their microphysical properties
(density, size, and shape), making accurate discrimination between rimed and aggregated particles particularly challenging
(Karrer et al., 2022; Planat et al., 2021; Thompson et al., 2014). To address this challenge, we develop an innovative
identification method using ground-based triple-frequency radar observations. Unlike the conventional threshold-based
approaches that rely on absolute radar measurements, our technique exploits the vertical- gradient signatures reflected from
multiple radar parameters. This approach provides a more robust physical framework by simultaneously tracking particle
size evolution, morphological changes, and dynamic processes during hydrometeor descent.

The paper is organized as follows: Section 2 describes the observational dataset and methodological framework, Section 3
presents a comparative analysis of both methods applied to a snowfall case, and Section 4 discusses the performance of new
method as well as its potential, and summarizes this study in Section 5.

## 2 Data and Methods

### 2.1 Data

The ground-based triple-frequency (X, Ka, W band) Doppler radar observations used in this study were collected from the
"TRIPle-frequency and Polarimetric radar Experiment for improving process observation of winter precipitation" (TRIPEx-
pol) field campaign(https://doi.org/10.5281/zenodo.1341389). The W-band radar is a frequency-modulated continuous-
wave (FMCW) system manufactured by Radiometer Physics GmbH (Küchler et al., 2017; Myagkov et al., 2020). The X-
and Ka-band radars are pulsed systems made by Metek GmbH(Görsdorf et al., 2015; Mróz et al., 2021), and both were
deployed in a vertically pointing (zenith) configuration, continuously recording Doppler spectra and echo power (Dias Neto
et al., 2019). The X-band radar (9.4 GHz) operates in a simultaneous transmit and receive (STAR) mode, measuring standard
polarimetric variables, with a sensitivity of –10 dBZ at 5 km and a vertical resolution of 30 m. The Ka-band radar (35.5 GHz)
transmits linearly polarized pulses and simultaneously receives co- and cross-polarized signals to derive LDR, offering a
sensitivity of –39 dBZ at 5 km and a vertical resolution of 28.8 m. The W-band radar (94 GHz) has no polarimetric
capability, with a sensitivity of –33 dBZ at 5 km and a vertical resolution of 16–34.1 m. Because the native vertical
resolutions of the three radars differ slightly, their data were re-sampled to a uniform vertical resolution of 30 m. All three
reflectivity datasets were corrected for atmospheric gaseous attenuation using the PAMTRA model (Maahn et al., 2015)
based on the gas absorption model of Rosenkranz (1993, 1998, 1999).

The TRIPEx-pol campaign was conducted by the German Weather Service (DWD) at the Jülich Observatory (Germany)
from November 2015 to February 2016, with the aim of comprehensive understanding of winter precipitation microphysics
through coordinated multi-frequency radar observations (Dias Neto et al., 2019). The high-quality multi-frequency radar
dataset from TRIPEx-pol provides unique advantages for winter precipitation research and supports several important studies.
For example, in terms of radar data quality control, Myagkov et al., (2020) used this dataset to improve multi-frequency
radar calibration methods; for precipitation microphysics retrieval, Mróz et al., (2020) developed algorithms for raindrop size





distribution parameters; regarding phase-transition processes, Mróz et al. (2020) and Karrer et al. (2022) performed in-depth analyses of melting-layer snow-to-rain conversion mechanisms. These efforts demonstrate the significant potential of TRIPEx-pol dataset in revealing cloud microphysical processes.

In this paper, we plan to use the triple-frequency radar observations provided by the TRIPEx-pol dataset to focus on
identifying the riming and aggregation processes in a typical snowfall event from January 3rd to 4th, 2016.

## 2.2 Riming and Aggregation Identification Methods

Here we adopt two methods to identify riming and aggregation during the snowfall event, more attention to the zone where ice particles grow actively extending from the melting layer up to an altitude where the temperature is about -15 °C (Chellini et al., 2022; Teisseire et al., 2025).

### 2.2.1 Multi-Parameter Threshold Method

Unlike previous studies that relied on a single parameter threshold or only 2–3 combined thresholds, here we utilize six radar parameters, including four observation parameters, such as, including Ze、MDV、SW and LDR from the Ka-band, and two derived parameters, DWR and volume-weighted mean diameter ($D_0$) (all Ze, MDV, SW, LDR values used are from the Ka-band). $D_0$ is retrieved iteratively based on the relationship between DWR and particle attenuation  (Gaussiat et al., 2003).

To eliminate the influence of tiny ice crystals in the initial stage of snowfall, we only analyze the periods when $D_0$ exceeds 2 mm, so as to focus on the particles for which the growth characteristics can be effectively observed. Table 1 lists the specific criteria for identifying riming and aggregation using the six parameters: if a given parameter meets the criterion for riming or aggregation, it contributes a score of 1. To ensure robust identification, a total score greater than 3 (i.e. satisfying more than half of the six criteria) is required to classify the process as riming or aggregation; otherwise, it is labeled as
"uncertain".

**Table 1: Criteria for identifying rimed and aggregated particles in the Multi-Parameter Threshold Method.**

| | Rimed particles | Aggregated particles | References |
|---|---|---|---|
| Ze(dBZ) | < 25 | > 25 | (Bringi et al., 2017; Kneifel et al., 2015) |
| MDV(m/s) | > -1 | < -1 | (Grazioli et al., 2015a; Kneifel and Moisseev, 2020; Mason et |
| SW(m/s) | > 0.5 | < 0.3 | al., 2018; Mosimann, 1995; Vogl et al., 2022) |
| LDR(dB) | > -20 | < -28 | (Bringi et al., 2017; Dias Neto et al., 2019; Teisseire et al., 2025; Tyynelä and Chandrasekar, 2014; Vogl et al., 2022) |
| $D_0$(mm) | > 3 | < 5 | (Braham, 1990; Bringi et al., 2017; Gaussiat et al., 2003; Teisseire et al., 2025; Von Terzi et al., 2022) |
| DWR(dB) | $DWR_{X-K_a} > 3$ <br> $DWR_{K_a-W} < 8$ | $DWR_{K_a-W} > 5$ | (Dias Neto et al., 2019; Kneifel et al., 2015; Mason et al., 2018; Tridon et al., 2022) |





### 2.2.2 Gradient-Based Multi-Parameter Identification Method

To further improve the flexibility and robustness of the identification, we introduce a multi-parameter vertical-gradient (VG)
identification method based on the PIVS approach proposed by Planat et al. (2021) for single-frequency radar. The newly
VG approach integrates the advantages of multiple parameters from the triple-frequency radar to achieve a
comprehensive discrimination of regions where riming and aggregation occur.

The vertical gradient of a radar variable is defined as $\nabla P = -\frac{\Delta P}{\Delta h}$, where $\Delta P$ is the change between two adjacent height levels
for the variable $P$ (e.g., DWR, MDV, SW, or LDR); $\Delta h$ is the vertical height difference (m); the negative sign represents the
gradient from top to bottom.

First of all, considering the importance of DWR for triple-frequency radar, not only the criteria for DWR in Table 1 is still
retained in this method, but also $\nabla_{DWR}$ is used for providing additional information. For example, if the X–Ka band DWR
increases downward ($\nabla_{DWR_{x-Ka}} > 0$) while the change in Ka–W band DWR is negligible, it indicates that during the falling
process, the particle size grows significantly, but the density is not increase proportional, which is a characteristic of
aggregation. In contrast, if the $DWR_{Ka-W}$ increases markedly with little change in the $DWR_{X-Ka}$, it implies that particles
become denser (without a large size increase) as they fall, a signature more consistent with riming. For Doppler velocity, a
negative vertical gradient ($\nabla_{MDV} < 0$, meaning fall speed increases with decreasing height) typically indicates the presence of
the riming process. As we know, ice particles become heavier and denser, leading to larger terminal fall velocities during the
riming process (Oue et al., 2021). Conversely, a positive gradient ($\nabla_{MDV} > 0$, slower fall speeds at lower altitudes) is mainly
observed in aggregation-dominated regions, since aggregation produces large, fluffy snowflakes that experience greater drag
and thus fall more slowly. $\nabla_{LDR}$ reflects changes in particle shape and orientation with height and is useful for distinguishing
growth modes: riming makes ice crystals more rounded and symmetric, causing LDR to decrease with descent ($\nabla_{MDV} < 0$),
whereas aggregation yields larger, irregular aggregates with increased non-sphericity, causing LDR to increase toward the
ground ($\nabla_{MDV} > 0$).
Based on the above analysis of vertical gradients, we establish threshold criteria for identifying riming and aggregation from
these gradient behaviors (Table 2).

**Table 2: Criteria for identifying riming and aggregation in the Gradient-Based Multi-Parameter Identification Method.**

|  | $\nabla_{DWR}$(dB/m) | $\nabla_{MDV}$(s⁻¹) | $\nabla_{SW}$(s⁻¹) | $\nabla_{LDR}$(dB/m) |
|---|---|---|---|---|
| Riming | $\nabla_{DWR_{x-Ka}} > 0$ | > 0 | > 0 | > 0 |
| Aggregation | $\nabla_{DWR_{Ka-W}} > 0$ | < 0 | < 0 | < 0 |



Similar to the multi-parameter threshold method, to ensure reliable classification we require a riming or aggregation total score > 2 (i.e. meeting more than half of the five criteria) to classify the process as riming or aggregation.

## 3 Results and Analysis

Figure 1 presents a snowfall event observed by the triple-frequency radar from 12:00 UTC on 3 January to 12:00 UTC on 4 January 2016. This event involved a variety of cloud and precipitation systems, including non-precipitating ice clouds,
stratiform snowfall, and shallow mixed-phase clouds(Dias Neto et al., 2019). Selecting such a snowfall case is beneficial for comprehensively evaluating the practicality and stability of the identification methods under complex weather conditions.

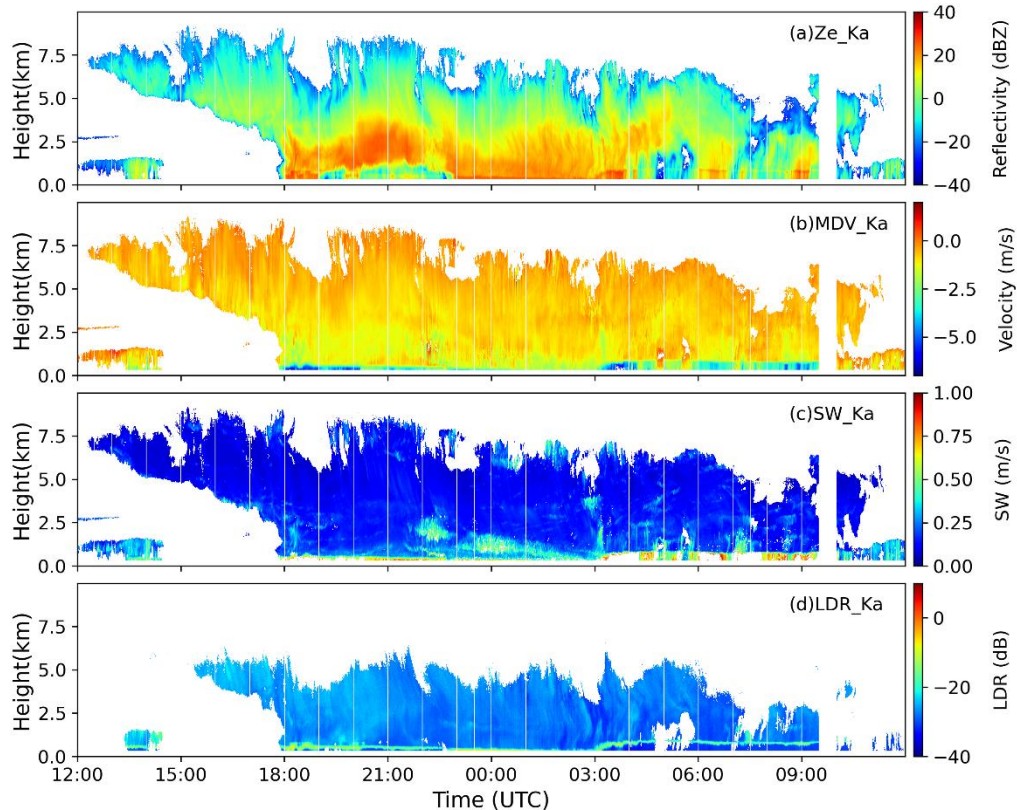

**Figure 2: Profiles of Ka-band radar Ze (Fig. 1a), MDV (Fig. 1b), SW (Fig. 1c), and LDR (Fig. 1d) during the snowfall event from 12:00 on 3 January 2016 to 12:00 on 4 January 2016.**

Before 03:00 UTC on 4 January, a typical melting-layer structure was identifiable around 0.5 km altitude, manifested as a pronounced horizontal reflectivity band accompanied by a sharp decrease in MDV, a broadening of SW below that layer, and an increase in LDR. These multi-parameter signatures confirm the existence of a distinct ice–liquid phase transition at this height (Brast and Markmann, 2020; Li et al., 2020; Romatschke, 2021). After 03:00, the melting layer rose to around





1 km. Similar abrupt changes were still observed in the Ze, MDV, SW, and LDR profiles at that level, indicating that a phase

transformation of particles occurs (i.e., the melting layer generated by the ice-to-liquid phase change).

The Ze profiles of Ka-band in Fig. 1a exhibit a clearly layered structure that evolves throughout the snowfall event. From 18:00 to 22:00 UTC on 3 January, the low-level cloud below 2 km showed relatively weak reflectivity (Ze < 10 dBZ), whereas a persistent strong echo band (peaking > 25 dBZ) appeared in the mid-level cloud between 2–4 km. After about 03:00 UTC on 4 January, the overall Ze became noticeably weaker. From the Ze vertical cross-sections and their evolution,

the active zones of different ice-phase growth processes can be preliminarily identified: the strong echo layer at a height of 2–4 km was likely associated with riming process, whereas the locally weaker echoes observed in the cloud at low altitude (< 2 km) after 03:00 on January 4 may be related to the in-cloud aggregation process.

The time–height evolution diagrams of the MDV (Fig. 1b), SW (Fig. 1c), and LDR (Fig. 1d) profiles further confirm the above identification of the riming and aggregation regions based on Ze. From 18:00 to 22:00 on January 3rd, at the altitude

of 2–4 km, the MDV values generally increased (with the maximum downward velocities approaching 3m/s), the SW exceeded 0.3 m/s, and LDR remained at a relatively low level overall. All of these characteristics suggest that riming-dominated microphysical processes were occurring in that region. In contrast, within the cloud near the melting layer, the microphysical processes were likely dominated by aggregation: the fall velocities became notably smaller (mostly less than 1 m/s), and LDR increased markedly, reaching above –15 dB in some areas.

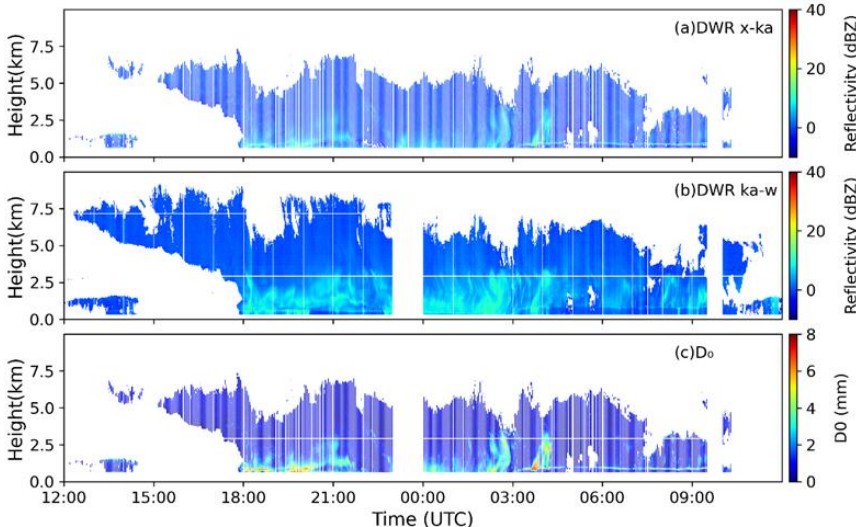


**Figure 2: Distributions of DWR (Fig. 2a and Fig. 2b) and D$_0$ (Fig. 2c) during the snowfall event.**

Figures 2a and 2b show two sets of DWR vertical cross-sections. It is evident that between 18:00 and 22:00 UTC on 3 January, in the mid-level cloud region (~2–4 km altitude) the following situation occurred repeatedly: DWR$_{Ka-W}$ increased significantly while DWR$_{X-Ka}$ remained relatively small. For example, around 20:00 UTC at 2–3 km height, DWR$_{Ka-W}$

reached approximately 8–10 dB, whereas the corresponding DWR$_{X-Ka}$ was below 3 dB. This DWR pattern implies that the





observed particle sizes produced a strong non-Rayleigh scattering effect at the higher-frequency W-band, but were not large

enough to depart from Rayleigh scattering at the lower-frequency Ka and X bands. This is exactly the typical triple-frequency radar signature of graupel formed by riming: $DWR_{X-K_a}$ remains relatively low while $DWR_{K_a-W}$ can reach on the order of 5–10 dB. Moreover, in this snowfall event, many strong-echo regions in the mid-level precipitating cloud (2–5 km)

displayed the similar features, indicating the prevalence of riming processes at those altitudes.

By contrast, during the same period from 18:00–22:00 UTC on 3 January, in the area below 2 km the radar observed a completely different dual-wavelength signatures. In the lower layer, $DWR_{X-K_a}$ was positive and both $DWR_{X-K_a}$ and $DWR_{K_a-W}$ increased almost simultaneously with nearly equal magnitude. This behavior is consistent with the scattering features of large, low-density, fluffy snowflakes. Moreover, when $DWR_{K_a-W}$ reaches saturation (around 7-10 dB),

$DWR_{X-K_a}$ continues to increase, resulting in a typical "hook-shaped" pattern for the relationship between $DWR_{X-K_a}$ and $DWR_{K_a-W}$ (Kneifel et al. 2015;Dias Neto et al. 2019). Similarly, the relatively weak echo bands at low altitudes in Fig. 1a–c reflects that aggregation processes were dominant. and the intermittent large peaks in $DWR_{X-K_a}$ further confirm that aggregation produced very large snowflakes.

The time evolution of $D_0$ in Fig. 2c also reflects the different microphysical processes within the cloud. At high altitudes of

6–9 km, $D_0$ was only a few hundred micrometers to ~1 mm, indicating that the initial ice crystals aloft were very small. As the particles fell into the mid-level (2–4 km), $D_0$ gradually increased to around 2.5 mm, meaning that the particles underwent significant growth in that height range. When the hydrometeors fell into the lower altitude (below ~2 km), $D_0$ reached its maximum value during this process, exceeding 4 mm in some periods. In addition, $D_0$ values observed within the 2 km is close to 2 mm, which is comparable to the size of typical graupel or dense snow grains. Aggregation process generates large,

loosely snowflakes ($D_0$ often > 3 mm); however, due to their low density and weak reflectivity, such aggregates manifest as extremely large $DWR_{X-w}$ . Overall, the vertical distribution of $D_0$ is consistent with the indications from the triple-frequency DWR and polarimetric parameters. During 18:00–22:00 UTC on 3 January, the high-altitude particles were small and light; the mid-level particles became denser through riming and grew moderately to 2–4 mm; and at the lower levels, the hydrometeors rapidly grew to their maximum size through via aggregation. By the early morning of January 4th, aggregation

had become the dominant process within cloud. Particles grew rapidly from about 4 km, and the growth rate was notably higher than that during the night of January 3rd.

In this snowfall event, the spatiotemporal evolution of Ze, MDV, SW, LDR, and DWR observed by the triple-frequency radar exhibited a high degree of consistency, and they jointly confirmed the dominant regions of different ice-particle growth mechanisms. A strong Ze, high fall velocity, broad SW, and extremely low LDR values together indicate that particles are

compact in structure and nearly spherical in shape; combined with the DWR characteristics (a relatively small $DWR_{X-K_a}$ and a markedly increased $DWR_{K_a-W}$), these features identify a typical riming-dominated region. In contrast, regions with moderately weak Ze, very slow fall speeds, an extremely narrow SW, and substantially elevated LDR correspond to more irregular, low-density particles. In those regions, $DWR_{X-K_a}$ and $DWR_{K_a-W}$ increase in unison (with the latter tending to





saturate at ~7 dB), consistent with the signature of an aggregation-dominated snow growth process. The consistency of these
radar variables at different stages and altitudes greatly enhances the physical interpretation of the particle growth processes,
providing reliable support for distinguishing the spatial distribution of aggregation and riming within the cloud.

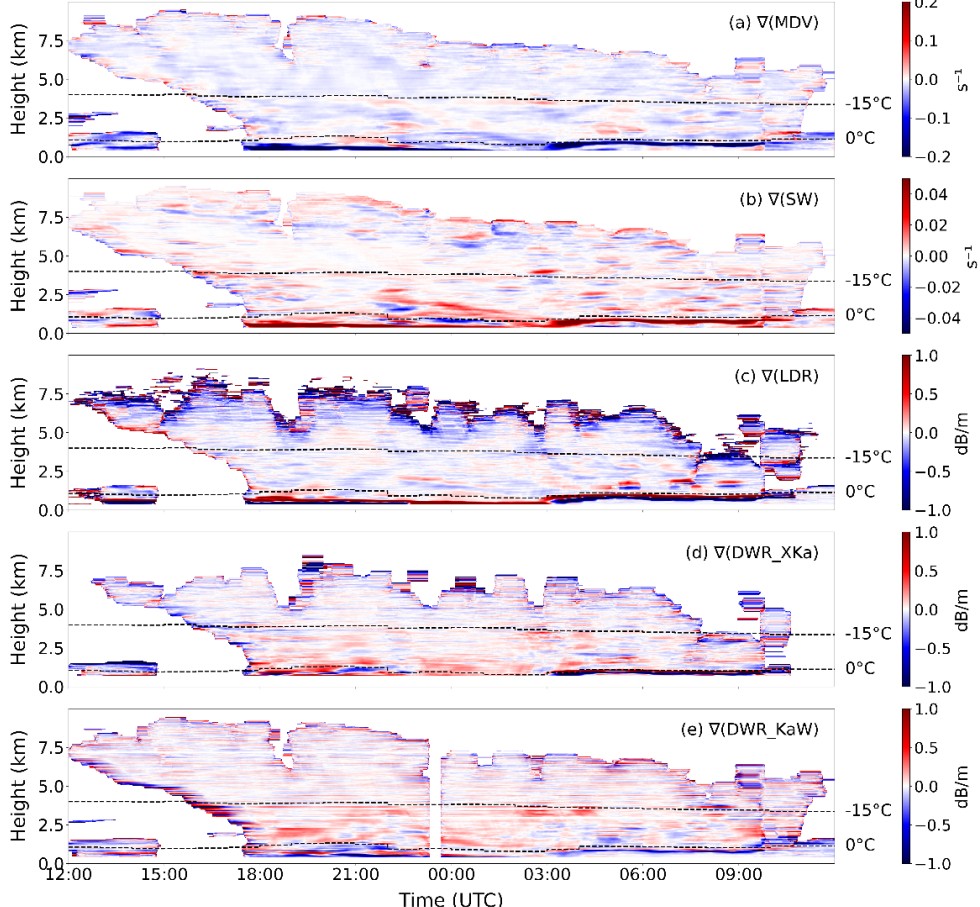

**Figure 3: Vertical gradient profiles of triple-frequency radar observations during the snowfall event, including MDV (Fig. 3a), SW (Fig. 3b), LDR (Fig. 3c), DWR$_{X-Ka}$ (Fig. 3d), and DWR$_{Ka-W}$ (Fig. 3e). The dashed lines indicate the 0 °C and −15 °C levels.**

Figure 3 shows the vertical profiles of calculated vertical gradients of various observables from the triple-frequency radar.
During 18:00–20:00 UTC on January 3 and 00:00–02:00 UTC on January 4, the vertical gradients reveal a pronounced
narrow band near the 0 °C level, with gradient signatures of $\nabla_{MDV} > 0$, $\nabla_{SW} < 0$, $\nabla_{LDR} > 0$. This series of gradient
characteristics indicate that at this height, the particle falling velocity slowed down, the SW narrowed, and particle shapes
became more irregular. Meanwhile, in that region DWR show $\nabla_{DWR_{x-Ka}} > 0$ and $\nabla_{DWR_{Ka-W}} < 0$, meaning the DWR$_{x-Ka}$
continued to increase but DWR$_{Ka-W}$ had begun to saturate and even slightly decrease. All of these gradient features indicate
that this height band was strongly dominated by aggregation processes, consistent with the analysis above. In the cloud
layers immediately above and below this aggregation-dominated band, the gradient sign of each parameter reverses ($\nabla_{MDV} <$




0、$\nabla_{SW} > 0$、$\nabla_{LDR} < 0$、$\nabla_{DWR_{x-K_a}} < 0$、$\nabla_{DWR_{K_a-W}} > 0$), suggesting that in those layers the particles' fall speeds increased, SW broadened, and shapes tended toward spherical. These are typical characteristics of a riming process, albeit of relatively

weaker intensity.

For 20:00–22:00 UTC on January 3, the overall vertical gradient in the lower-middle cloud (1–2.5 km) was $\nabla_{MDV} < 0$、$\nabla_{SW} > 0$、$\nabla_{LDR} < 0$. This indicates that during this period, within that height range, particle fall speeds increased, SW broadened, and shapes became more spherical, with riming being the dominant process. However, two thin layers around ~2.5 km and the 0 °C level exhibited the opposite gradient pattern, indicating that within these narrow layers the fall speeds slowed, SW

decreased, and particle shapes grew more irregular—signaling the onset of snowflake aggregation. This shows that even during an overall riming-dominated stage, an initial aggregation of ice crystals can occur at certain local heights.

During 01:00–03:00 UTC on January 4, the vertical gradients in the lower-middle cloud (1–4 km) were $\nabla_{MDV} < 0$、$\nabla_{SW} > 0$、$\nabla_{LDR} < 0$、$\nabla_{DWR_{x-K_a}} > 0$, $\nabla_{DWR_{K_a-W}} > 0$. 这 This means that the region exhibited typical riming characteristics—such as accelerated fall speeds and particle shapes becoming more spherical—and, moreover, the positive gradient in $\nabla_{DWR_{x-K_a}} > 0$

indicates that particle sizes had grown significantly compared to the initial ice crystals: an ice shell formed on the surface of the rimed particles, substantially increasing their diameter and indicating strong riming. Between 03:00–05:00 UTC, the cloud's vertical gradient characteristics changed dramatically, with a broad region showing $\nabla_{MDV} > 0$、$\nabla_{SW} < 0$、$\nabla_{LDR} > 0$、$\nabla_{DWR_{x-K_a}} > 0$, and $\nabla_{DWR_{K_a-W}} < 0$. All of these features indicate that aggregation became the dominant process in those areas. Notably, during this period a distinct vertical stratification emerged: at around 2 km altitude, a layer still maintained gradient

signatures consistent with the earlier riming process, meaning that even in an aggregation-dominated stage, riming was still ongoing at ~2 km in the cloud. This vertical layered structure demonstrates that during the evolution of precipitation, microphysical processes at different altitudes can coexist — the growth of snow crystals by aggregation occurs at most heights, while a certain degree of riming continues locally.

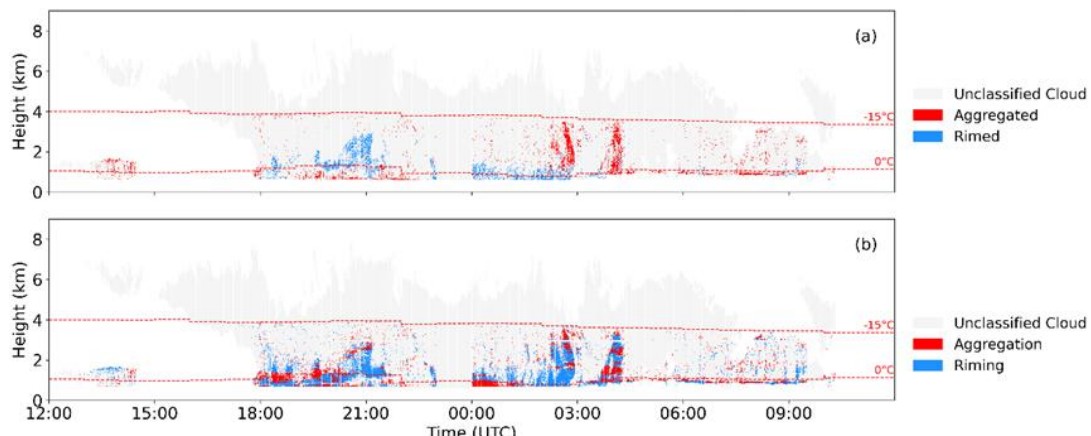

**Figure 4: Rimed and aggregated particles and their regions of occurrence identified by the two methods (a: Multi-Parameter Threshold Method; b: Gradient-Based Multi-Parameter Identification Method).**

 

Using multiple radar observables and their vertical gradient data, we applied the two methods introduced earlier to identify the riming and aggregation processes during this snowfall event, as shown in Fig. 4. The identification results from both methods are generally consistent in terms of spatial distribution and evolution trends. Specifically, during the early-to-middle

stage of the snowfall (18:00 UTC Jan 3 to 01:00 UTC Jan 4) in the low-level cloud region, and in the later stage (01:00– 04:00 UTC Jan 4) throughout the entire lower and middle cloud layers, both methods identified pronounced aggregation processes. In the mid-stage of the snowfall (21:00 UTC Jan 3 to 01:00 UTC Jan 4) within the lower–mid cloud, both methods successfully captured a region dominated by riming.

Although the two identification methods are generally consistent in discerning the macro-scale trends, they still exhibit

significant differences in identification accuracy and sensitivity at the boundaries of microphysical process regions and during transition stages. To further analyze the causes of these differences and to verify their physical soundness, we selected four representative time–height intervals of the snowfall in which the identification results diverged markedly, and plotted two-dimensional scatter distributions of $DWR_{x-K_a}$ vs. $DWR_{K_a-W}$ (Fig. 5), with different colors indicating the frequency of occurrence of particles. We also superimposed the classical "triple-frequency radar hook" curves (blue solid line

representing a typical riming process, red solid line a typical aggregation process; curves from Kneifel et al. 2015) to evaluate the accuracy of the identification results from both methods. The specific analyses are as follows:

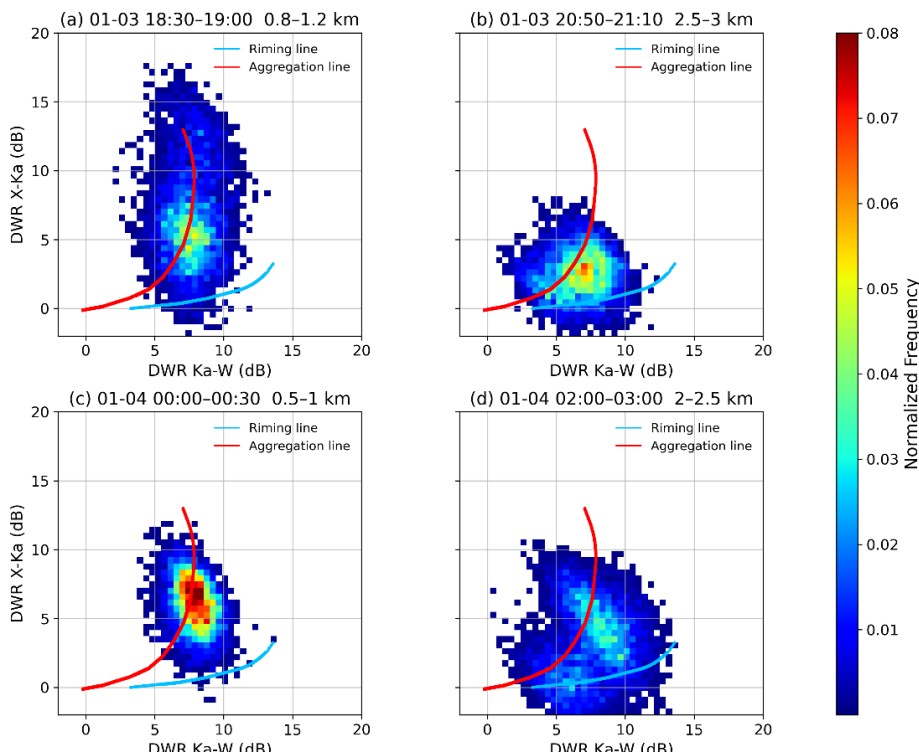

Figure 5: Scatter plots of DWRX-Ka vs. DWRKa-W for four representative time–height intervals. The blue solid line represents a typical riming process, and the red solid line represents a typical aggregation process (curves from Kneifel et al. 2015).



For the intervals 18:30–19:00 UTC on January 3 at 0.8–1.2 km (Fig. 5a) and 00:00–00:30 UTC on January 4 at 0.5–1 km (Fig. 5c), the Multi-Parameter Threshold Method identified both as riming-dominated, whereas the Gradient-Based Multi-Parameter Identification Method identified them as aggregation-dominated. The scatter plots show that the observed points are distributed mainly near the theoretical aggregation curve, with the high-concentration region clearly extending along the aggregation curve. This distribution pattern strongly indicates that these regions were dominated by aggregation,

corroborating the accuracy of the gradient-based identification. Notably, the scatter distribution in Fig. 5a extends further toward higher $DWR_{X–Ka}$ values compared to that in Fig. 5c, indicating a more intense degree of aggregation. This is consistent with the observation that $D_0$ was significantly larger during that period (Fig. 2c), further supporting the physical mechanism by which aggregation produces large, low-density snowflakes.

For the period 20:50–21:10 UTC on January 3 at 2.5–3 km altitude (Fig. 5b), the Multi-Parameter Threshold Method

identified this region as riming-dominated, whereas the Gradient-Based Multi-Parameter Identification Method identified it as aggregation-dominated with local riming characteristics. The scatter plot shows that the observations fall near both the theoretical aggregation and riming curves, exhibiting a distinct bimodal distribution; however, the high-frequency concentration leans more towards the aggregation curve. This distribution pattern indicates the co-existence of riming and aggregation processes in this region, with aggregation being relatively dominant. By analyzing the continuous vertical

variations of the observables, the Gradient-Based Multi-Parameter Identification Method was able to more sensitively capture this co-existence signal.

For the period 02:00–03:00 UTC on January 4 at 2–2.5 km (Fig. 5d), the Multi-Parameter Threshold Method identified this layer as aggregation-dominated, whereas the Gradient-Based Multi-Parameter Identification Method identified it as a region of coexisting aggregation and riming. Considering the preceding and subsequent periods, this stage can be inferred to be a

transition from riming to aggregation. The scatter plot shows a transitional pattern from the riming curve toward the aggregation curve. The observation points are broadly distributed between the two theoretical curves, and the high-concentration region extends along a "transition band" connecting the riming and aggregation curves. In particular, the $DWR_{Ka–W}$ remains at a moderately elevated level (peaking at ~13 dB) and has not yet reached the saturation level of the aggregation process, indicating that riming still plays an important role during this period. This dynamic transitional feature

underscores the superior ability of the Gradient-Based method to identify microphysical process transitions, as it can capture the continuous details of process evolution.

Through the above comparative analysis of the four typical differing regions, it is evident that the Gradient-Based Multi-Parameter Identification Method exhibits stronger capabilities than the threshold-based method in the following respects: It provides a more detailed characterization of regions with mixed processes, avoiding the oversimplification of microphysical

processes into a single dominant process. It enables earlier identification of aggregation processes with weak signals, capturing process indicators through trend changes even before aggregation features become prominent. It captures the evolution during process transition stages, identifying the continuous evolution of microphysical processes rather than treating them as discrete states. These advantages stem from the principle of the Gradient-Based Multi-Parameter



Identification Method, which analyzes relative changes rather than absolute values. This makes it more adaptable to systematic factors such as measurement errors and calibration biases, and more consistent with the continuous evolution of microphysical processes in the atmosphere

## 2 Discussion and Conclusion

In this study, ground-based triple-frequency radar observations were used to identify and compare riming and aggregation microphysical processes in a snowfall event, using both the Multi-Parameter Threshold Method and the Gradient-Based Multi-Parameter Identification Method. Both methods are fundamentally parameter-based approaches, and overall each was able to effectively reveal the primary microphysical characteristics of this snowfall event (coexisting riming and aggregation in the early stage, riming-dominated in the middle stage, and aggregation-dominated in the later stage), indicating a certain consistency in capturing the macroscopic evolution of the event. However, in the boundary regions and during the transition stages of microphysical process, clear differences emerged between the two. The Multi-Parameter Threshold Method, relying on preset absolute thresholds, has strong skill in identifying pronounced, well-defined process features, but when confronted with complex structural details such as riming-to-aggregation transitions or mixed-phase regions, it can miss or misclassify processes – lacking generality across different scenarios and the ability to resolve fine-scale processes. In contrast, by analyzing the relative vertical variation trends of multiple parameters, the Gradient-Based Multi-Parameter Identification Method can more finely depict the evolution of particles during their descent. In particular, when the upward shift of the DWR scatter plot exhibits a hook-shaped distribution, indicating enhanced aggregation, the vertical-gradient approach demonstrates a stronger ability to discern this strengthening signal. In this snowfall case, this method yielded an identification field that best aligned with the cloud's microphysical evolution, encompassing the main trends while also capturing process transitions and coexisting features within the complex cloud.

Overall, the Gradient-Based Multi-Parameter Identification Method we proposed in this paper improves the triple-frequency radar's capacity to capture the spatial structure and transition details of microphysical processes, while preserving the quantitative nature of the radar observations. This process-oriented identification approach avoids relying on absolute thresholds. Instead, it captures the particle evolution signals through relative changes, thus effectively reducing the impact of environmental factors such as inter-system calibration errors, propagation attenuation, and seasonal or site differences. This shift from "observing phenomena" to "understanding changes" in analysis is more suitable for dissecting the evolution of microphysical processes in complex, dynamically evolving precipitation clouds. The framework of this method provides strong support for establishing a precipitation microphysical classification system with stronger physical consistency and broad applicability across scenarios. It has positive significance for further improving the parameterization schemes of ice-phase process and enhancing the accuracy of snowfall forecasts.




*Author contributions.* WD carried out the investigations and wrote the manuscript. HW, BY, XX and CH contributed input and advise at all stages of the scientific discussions and of the manuscript writing. All co-authors revised and commented on the manuscript.

*Competing interests.* The authors declare that they have no conflict of interest.

*Acknowledgements.* The authors gratefully acknowledge the TRIple-frequency and Polarimetric radar Experiment (TRIPEX-pol) for providing the high-quality radar dataset used in this study. These observations were essential for the identification and analysis of microphysical processes associated with winter precipitation. We also extend our appreciation to the
TRIPEX-pol team for their dedicated efforts in field deployment, instrument operation, and data management. This work was supported by the National Key Research and Development Program of China (Grant 2022YFF0801301) and the Innovation Foundation of CPML/CMA (Grant 2023CPML-A01).



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
