# Peer review of "Identification of Snowfall Riming and Aggregation Processes Using Ground-Based Triple-Frequency Radar"

_EGUsphere, 2025_

## Author Comment (AC2)

**Reviewer Comment 2、3:** *"Line 180-182: why do you say riming? I don't see a MDV increase here at all. Do you have any other indications? Or is this solely based on enhanced Ze? I don't agree with the statement that the lower Ze regions are aggregation, you can have similar Ze values with riming and aggregation, that's why most previous studies distinguish riming with the MDV. In my opinion, the region of low Ze could be connected to sublimation processes, or something else.*

5 *Aggregation usually increases Ze. I would suggest you adapt the MDV colorscale to show the expected values in snow better, perhaps until -4m/s, not until -7.5m/s. Also adapt your LDR colorscale, it is very hard to see anything, since most values are below -20"* *"Line 188: how do you expect rimed particles to slow down due to aggregation again? I would much rather say that sublimation plays a role here. Especially because Ze decreases. If you had aggregation of the previously rimed particles I would Ze to continuously increase (due to the size increase), or at least stay constant."*

10 **Response:** Thank you for the suggestion. We have adjusted the color scale accordingly and re-examined the description after 03:00 UTC. Our original interpretation—attributing the post-03:00 UTC low-level echo weakening solely to aggregation—was incomplete. In reality, once the snowfall entered a drier layer, the smaller ice crystals likely sublimated first, leaving only the larger snowflakes; this resulted in reduced reflectivity and an increased DWR. This behavior is opposite to what one would expect from pure aggregation: if aggregation alone were producing larger snowflakes, the reflectivity should increase

15 or at least not decrease. Therefore, we introduced the sublimation process to explain the weak low-level echoes, which more accurately reflects the physical processes at that time. We have analyzed Ze together with MDV and SW to illustrate the role of sublimation in this stage, and we have revised the text as follows: *"Line 180-182 on Page 8: After about 03:00 UTC on 4 January, the overall Ze weakened substantially, with only very faint echoes remaining at low altitudes. At the same time, the MDV increased toward about –1 m/s, the SW broadened further (exceeding 0.3 m/s). These combined signatures suggest that*

20 *by this time, riming and pure aggregation were no longer the dominant processes in the cloud. Instead, the presence of large, low-density snowflakes (along with the sharp drop in reflectivity) indicates that many of the smaller ice particles were likely undergoing sublimation (partial or complete evaporation) as they fell through a drier layer of air."* .

[Figure]

**Reviewer Comment 5:** *"Line 203: this behaviour could also be consistent with sublimation, as the smallest particles are expected to be sublimated faster than the larger ones, therefore shifting D0 towards larger sizes and increasing DWRKaW and DWRXKa. I would say it is more likely that feature because Ze decreases."*

**Response:** Thank you for the suggestion. Indeed, between 20:00 and 22:00 UTC, there was a sublimation process that we had previously overlooked. In the figure below, we have marked the specific region with a red dashed outline and arrow. We have also corrected the text in the manuscript accordingly: "In 206 line on Page 9: *Between 20:30–22:00 UTC on 3 January, around the 1 km altitude level, the radar observations reveal signatures distinct from those at higher altitudes. In this near-surface layer, the Ze decreases noticeably with decreasing height, indicating substantial particle loss. Meanwhile, the MDV becomes less negative (rising to values above –1 m/s). However, the $DWR_{Ka-W}$ remains significantly positive. This combination – a marked reduction in Ze, a pronounced increase in MDV, and a sustained high DWR – is a distinctive radar signature of snow-particle sublimation."* .

[Figure]

35

**Reviewer Comment 9:** *"In your Section 2.2.2 I am missing a citation of Kumjian et al. 2022, and references therein, they have done significant work in identifying fingerprints of ice microphysical processes by studying the gradients of radar*

40 *variables."*

**Response:** Thank you for the suggestion. We initially did not cite Kumjian et al. (2022) in our methods section because that study focuses on $Z_H$, $Z_{DR}$, and $K_{DP}$, which do not directly align with our criteria.

| Microphysical Processes | $\Delta Z_H$ | $\Delta Z_{DR}$ | $\Delta K_{DP}$ |
|---|---|---|---|
| Collision-Coalescence | + | + | + |
| Breakup | − | − | − |
| Evaporation | − | +[−] | − |
| Size Sorting | − | − | + |
| Vapor Deposition | + | + | + |
| Aggregation | + | − | − |
| Riming | + | − | − |
| Riming with ice splintering | + | − | + |
| Sublimation | − | − | − |
| Sublimation with fragmentation | − | − | + |
| Refreezing | − | +[−] | +[−] |
| Melting * | + | + | + |

However, this paper indeed provides important insights into using vertical gradients as "fingerprints" of ice-phase microphysical processes and has greatly inspired our work. We will add a citation to Kumjian et al. (2022) in the Introduction: "In 73 line on Page 2: *Kumjian et al. (2022) provide a comprehensive review illustrating that vertical gradients of dual-polarization radar observables serve as distinctive "fingerprints" of precipitation microphysical processes, including those in the ice phase.*".

**Reviewer Comment 10:** *"Line 170: is this basically at the lowest range gate? I find it very hard to see the melting layer between approx."*

**Response:** Yes, this refers to the gates near the lowest range bin of the radar. Between ~22:30 UTC and 03:00 UTC, the melting layer becomes more discernible when examining the LDR profile. We adjusted the color scale and use a red box in the figure as shown here to highlight the melting layer, making it easier to identify.

[Figure]

In 168 line on Page 7.

**Reviewer Comment 12:** *"Figure 2: why is your colorbar limit of DWR KaW so high? Also, on your colorbar you are stating Relfectivity. What is it then? Reflectivity or DWR? If it is DWR we do not expect DWRs to be larger than 15dB in most cases, and even that is already an extreme case. So I would suggest you change our colorbar to reflect the limits of DWR better."*

**Response:** Thank you very much for pointing out this issue. We will correct the color bar labeling and adjust the upper limit of the DWR color scale to a more appropriate value that reflects the typical range of DWR.

[Figure]

In 190 line on Page 8.

---

## Author Comment (AC3)

**Reviewer Comment 5:** *"discuss what temporal averaging was applied to the data and how the vertical gradient was computed (I suggest using Savitzky-Golay method)"*

Response: We thank the reviewer for this practical and constructive suggestion. We agree that clearly documenting the temporal averaging applied to the radar data and the procedure for computing the vertical gradients is important for the transparency and reproducibility of the method. In particular, based on the reviewer's suggestion, we have incorporated a Savitzky–Golay filtering approach into our data processing. The radar time–height data are smoothed in time (over a short moving window of a few minutes, similar to Planat et al., 2021) and in the vertical (using a 3-gate window) using a second-order Savitzky–Golay filter. These clarifications have been added to Section 2.2.2 of the revised manuscript "*Before computing the vertical gradients, we apply temporal averaging and vertical smoothing to the radar data to reduce small-scale noise while preserving the underlying microphysical signal. Specifically, following Planat et al. (2021), we average each radar profile with its neighboring profiles over a 10 minutes time window to filter out high-frequency fluctuations. Next, we smooth each profile in the vertical using a three-gate moving window (90 m) to reduce gate-to-gate noise. To implement these smoothing steps effectively, we employ a Savitzky–Golay (SG) filter in both time and height dimensions, fitting a second-order polynomial within the chosen window in each dimension. This SG-based smoothing approach preserves the shape of the vertical profiles while suppressing random noise, thus providing robust estimates of the gradients.*"

We have also re-plotted the relevant figures using the improved smoothing scheme and included them in the revised submission.

[Figure]

20